# Efficacy of Pendimethalin Rates on Barnyard Grass (*Echinochloa crus-galli* (L.) Beauv) and Their Effect on Photosynthetic Performance in Rice

Chinaza B. Onwuchekwa-Henry [1,2,*], Robert Coe [2], Floris Van Ogtrop [1], Rose Roche [2] and Daniel K. Y. Tan [1]

1   Faculty of Science, School of Life and Environmental Sciences, Sydney Institute of Agriculture, The University of Sydney, Sydney 2006, Australia
2   Agriculture and Food/DATA61, Commonwealth Scientific and Industrial Research Organisation (CSIRO), Canberra 2601, Australia
*   Correspondence: chinaza.onwuchekwa-henry@sydney.edu.au

**Abstract:** Pendimethalin herbicide toxicity to rice plants and barnyard grass invasion have increasingly affected the productivity of direct-seeded rice (DSR) in the fields. Whether and how to promote DSR productivity and sustain weed management depend on the appropriate pre-emergence herbicide application rate to minimise its toxicity in the rice ecosystem. Pot experiments were conducted to determine the effects of pendimethalin rates (1.5, 1.75, 2.0 kg a.i. ha$^{-1}$, two control treatments include the untreated control and the treated control with 1.5 kg a.i. ha$^{-1}$ *S*-metolachlor) on barnyard grass (*Echinochloa crus-galli* (L.) Beaux) and their potential toxicity risk to photosynthetic performances of rice (*Topaz* and *Sen pidao*). All the pendimethalin treatments provided excellent control of barnyard grass. Among the treatments, 1.5, 1.75, 2.0 kg a.i. ha$^{-1}$ pendimethalin and 1.5 kg a.i. ha$^{-1}$ *S*-metolachlor (treated control) decreased leaf area of barnyard grass significantly by 38.9, 49.6, 49.6 and 46.2%, respectively, compared with the untreated control at 40 days after sowing (DAS). The above-ground biomass of barnyard grass significantly decreased by 40% (1.48 g plant$^{-1}$) with 2.0 kg a.i. ha$^{-1}$ pendimethalin and by 46.2% (1.33 g plant$^{-1}$) when 1.5 kg a.i. ha$^{-1}$ *S*-metolachlor was applied at 40 DAS compared with the untreated pots. Higher pendimethalin rates increased toxicity in *Topaz* and *Sen pidao* varieties, and 2.0 kg a.i. ha$^{-1}$ pendimethalin significantly reduced effective quantum yield (light-adapted) of photosystem (PS) II by 18% (0.58) and 19% (0.52), respectively, compared with the untreated control. Application of 2.0 kg a.i. ha$^{-1}$ pendimethalin rate significantly decreased the maximum quantum yield (dark-adapted) of *Sen pidao* (0.66) compared with 1.5 kg a.i. ha$^{-1}$ pendimethalin (0.68) including the untreated control. All pendimethalin treatments suppressed above-ground biomass at 55 DAS, but above-ground biomass of barnyard grass significantly decreased by 59.9% when 2.0 kg a.i. ha$^{-1}$ pendimethalin was applied compared with the untreated control. Although application of 1.5 kg a.i. ha$^{-1}$ pendimethalin rates reduced the effective quantum yield (light-adapted) of photosystem II of *Sen pidao* (0.55) by a small percentage (8%) than *Topaz* (0.65), it was non-toxic for both varieties compared with 2.0 kg a.i. ha$^{-1}$ pendimethalin. Therefore, the use of 1.5 kg a.i. ha$^{-1}$ pendimethalin can be used for effective weed management in the direct seeding of rice at an early growth stage.

**Keywords:** pendimethalin; *S*-metolachlor; phytotoxicity; photosystem II; barnyard grass; rice varieties

## 1. Introduction

Barnyard grass (*Echinochloa crus-galli* (L.) Beaux) is one of the most troublesome weeds with rapid germination, high adaptability and competitive ability for limited resources such as light, water and nutrients in the lowland rice ecosystem [1]. Competition from barnyard grass occurs from the emergence of rice seedlings and continues up to the maturity stage which can lead to the removal of between 60 and 80% of available nitrogen (N) from the soil [2]. Barnyard grass has remarkable similarities with rice in morphology during growth,

which makes it difficult to control during the early vegetative phase in direct-seeded rice (DSR) systems [3,4]. Poor weed control management practices negatively impacted on yield in the rice-growing regions in Asia such as Cambodia and India [5,6]. For example, frequent usage of post-emergence herbicides to control weeds can potentially result in high contamination of barnyard grass weeds in harvested seed in lowland rice ecosystems [7]. The application of post-emergence herbicides has not been effective to suppress the growth of barnyard grass (*Echinochloa crus-galli*) in rice fields at the early growth because of fast competition from emergence, and its reliance has high potential for developing herbicide resistance in rice production systems [8].

Pendimethalin is a selective pre-emergence herbicide that has shown potential to suppress barnyard grass and other weed species at the early growth stage to maximise DSR yield [9]. Pendimethalin is a dinitroaniline compound that has the ability to impede microtubulin synthesis which is responsible for forming cell walls microfibrils that prevent cell enlargement and chromosome movement during mitosis [10]. Other pre-emergence herbicides, such *S*-metolachlor, can interfere with the protein synthesis and primarily be taken in by the germinating shoot and root [11]. *S*-metolachlor belongs to the chloroacetanilide group of herbicides with a low risk of developing weed resistance, which has been applied since the 1970s to more than 70 different crops [12]. A previous study has reported that pendimethalin exposure to plants potentially resulted in the reduction in the photosynthetic rate and efficiency of carboxylation in rice plants [13]. To reduce herbicide toxicity or oxidative damage to rice, it is critical that the right pendimethalin rate is applied to achieve sustainable barnyard grass management and DSR productivity. Several studies in DSR found varying results with different pendimethalin rates including other pre-emergence herbicides for weed control under different environmental conditions [14–16]. For instance, application of 1.6 kg ai ha$^{-1}$ pendimethalin caused injury on rice seedlings and reduced yield [17]. A different study reported that 2 kg ai ha$^{-1}$ pendimethalin exhibited high toxic symptoms in rice plants and reduced the shoot biomass by 21% under saturated soil–water conditions [18]. Furthermore, using pendimethalin rates between 0.6 and 1.1 kg a.i. ha$^{-1}$ increased seed mortality and reduced rice germination in DSR under aerobic conditions [19]. This study could be improved if a new source of data that captures the impact of different herbicide rates on the photosynthesis performance in rice is investigated.

Photosynthesis is key to crop yield, and the use of chlorophyll (Chl) fluorescence is extensively used in phenotyping to ascertain the risks that relate to herbicide toxicity and photo inhibition of photosystem II (PS II) of the plant during growth [20,21]. The physiological variable, such as effective quantum efficiency (Fv/Fm ratio), is a useful and critical parameter that can detect and monitor the photo inhibition induced by any stress factor in plants [22–24]. The use of a low pendimethalin rate produced a desired effect on barnyard grass and photosynthetic performance of photosystem II of rice varieties [25]; however, this information regarding pendimethalin-based weed management in DSR production systems is unpublished. In addition, single-photon avalanche diode (SPAD) is used as a proxy for relative chlorophyll content to monitor leaf greenness in relation to plant photosynthetic performance under abiotic stresses including herbicide damage. Despite the importance of pendimethalin herbicide to reduce barnyard grass in direct seeding, information on its toxicity to rice plants based on the application rates remain scarce. Therefore, investigating the photosynthetic performances in rice plants from herbicide toxicity would provide a basis to enhance and optimise pendimethalin for sustainable weed management and DSR productivity in lowland ecosystems. On this basis, we hypothesised that low pendimethalin rates decreased the biomass of barnyard grass and increased photosynthetic performances of rice varieties compared to high pendimethalin rates. Therefore, the specific objectives of the study were to: (1) assess the effectiveness of pendimethalin rates in controlling barnyard grass and (2) characterise rice varietal photosynthetic responses to pendimethalin rates.

## 2. Materials and Methods

### 2.1. Planting and Experimental Layouts

Different pot experiments were conducted for barnyard grass and rice seeds in the greenhouse (Cropatron) at the CSIRO-Australian Plant Phenomics Facility (APPF) in Canberra. The seeds of barnyard grass and rice varieties, *Topaz* (Australian variety) and *Sen pidao* (Cambodian variety), were obtained from Charles Sturt University and the Department of Primary Industries (DPI) New South Wales, Australia, respectively. All the experiments were conducted in an iron steel greenhouse with all the sides covered with a transparent glass of double-sealed pane that has up to 80–91% light transmission. The soil used for the barnyard grass and rice experiments comprised sterilised vertisols that contained added fertiliser of 1 g L$^{-1}$ of Aboska® with 14.2% N, 6.4% P, 5.1% K and 3 g L$^{-1}$ of calcium carbonate lime and soil pH of 6.5. Sterilised soils were filled in 50 mm rounded-plastic pots with holes underneath. The filled pots were placed in trays containing up to 1.5 L of water, and soils were wetted with standing water left in the trays to enhance seed germination. The pots were placed in a glasshouse under controlled conditions at 60% relative humidity and 35 and 25 °C day and night temperatures, respectively.

### 2.2. Weed and Rice Seeds Germination

For the barnyard grass experiment, each tray consisted of 12 pots arranged in a completely randomised design (CRD) with three replicates, producing 180 experimental units. Ten seeds of barnyard grass were sown in individual pots and different concentrations of pendimethalin at 1.5, 1.75 and 2.0 kg a.i. ha$^{-1}$ and two control treatments (untreated control and treated control as 1.5 kg a.i. ha$^{-1}$ *S*-metolachlor) were sprayed three days after sowing (DAS) for comparison according to treatment schedule. All herbicides were applied using a knapsack sprayer that delivered 200 L ha$^{-1}$ of spray solution at a spray pressure of 120 kilopascals (kPa) except the non-treated control.

For rice experiments, direct sowing and transplanting methods were used to collect plant information of *Topaz* and *Sen pidao* varieties. Direct sowing method was necessary to eliminate the challenge of poor seedling establishment and herbicide effects on rice germination under different growth media (Figure 1).

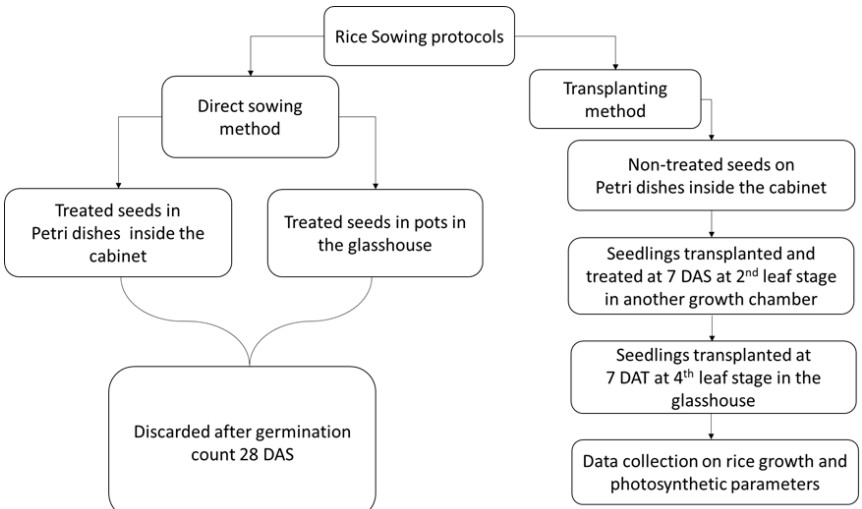

**Figure 1.** Different rice germination protocols for the experiments.

In direct sowing, twelve seeds of each rice variety were separately sown on Petri dishes and in pots and sprayed at different concentrations of 1.5, 17.5, 2.0 kg a.i. ha$^{-1}$ and two control treatments (untreated control and treated control as 1.5 kg a.i. ha$^{-1}$ *S*-metolachlor). Each experiment was arranged in 5 × 2 factorial in a complete randomised block design with three replicates. For Petri dishes, seeds were placed directly in 9 cm diameter on an evenly dampened filter paper with pendimethalin herbicide solution of

500 μL in 50 mL of water according to the treatment rates as described above. The seeds in Petri dishes and pots were placed in the growth cabinet and glass house, respectively. The sides of the Petri dishes were sealed with parafilm to prevent loss of moisture and placed in (incubator) cabinet at 60% relative humidity and 28/24 °C day/night temperatures with a photoperiod set at 16 h. Germination counts were recorded at 7 DAS when the radicles were approximately 2 mm in length.

In the transplanting method, seeds of each rice variety (*Topaz* and *Sen pidao*) were sown on Petri dishes and placed in an incubator at 60% relative humidity and 28/24 °C day/night temperatures set at a 16 h photoperiod. Then, rice seedlings were first transferred into 290 × 350 mm trays containing a mixture of soils (vertisols) and Osmocote® (NPK 15:2:13) after seven days in a growth chamber when the first true-leaf emerged, radicle and the coleoptile became visible and allowed for another 7 days. Then, rice seedlings of each variety were transplanted in the pots that contained mixtures of vertisols, rice mix and rice husks in the greenhouse at the fourth leaf stage. Each variety was sprayed using the same pendimethalin rates as described above. The pots were arranged in 5 × 2 factorial in a complete block design with six replications. Each block had six pots giving a total of 160 experimental units for each variety based on a combination of two factors in a factorial arrangement. Rice seedlings were allowed to grow for 14 days after treatment (DAT) of pendimethalin to determine herbicide toxicity in rice plants.

### 2.3. Growth Attributes on Weed and Rice

Data on germination counts of barnyard grass that survived were recorded. The total leaf area was measured with LICOR 3000C 40 and 55 DAS, and dry leaf and dry shoot biomass of the weed was measured at 40 and 55 DAS, respectively. The root biomass was collected at 68 DAS by carefully removing plants and washing off the soil.

Data on rice germination counts were only collected from the direct sowing methods conducted on Petri dishes in the cabinet. The total leaf area, leaf and shoot biomass of rice were collected 14 days after herbicide treatment (DAT) from the transplanting method in the greenhouse. Leaves and shoots of rice and weeds as well as roots biomass of weeds were placed in separate paper bags and oven-dried at 70 °C to determine constant dry biomass.

### 2.4. Chlorophyll Fluorescence and SPAD Values of Rice Varieties

Chlorophyll fluorescence of maximum quantum yield (Fv/Fm-dark-adapted) and effective quantum yield (F'v/F'm-light-adapted) of PS II were measured on three flag leaves from each plant at 14 DAT. Data on maximum quantum yield (Fv/Fm) were collected after dark adapting the leaves with plastic covers acclimated for 30 min at weakly modulated irradiation of $< 0.1$ μmol m$^{-2}$ s$^{-1}$ using Fv/Fm ratio using a portable plant fluorometer. The F'v/F'm was determined using MultispeQ that is a non-invasive portable instrument that measures the productivity health of plants related to photosynthetic parameters based on light-driven fluorescence yield (https://help.photosynq.com/instruments/multispeq-v2.0.html: accessed on 2 February 2023), where $F_v$ is the variable fluorescence yield, and $F_m$ means maximum chlorophyll fluorescence in the dark-adapted state. F'v represents the variable fluorescence yield, F'm is maximum chlorophyll fluorescence and $F_0$ means in the light-adapted state. SPAD data, which are proxy to relative chlorophyll content, were collected from the remaining three plants in each treatment block.

### 2.5. Statistical Analysis

All the data collected were analysed separately based on the treatments scheduled for rice and weed experiments. Rice was subjected to two-way ANOVA using pendimethalin rate and rice variety as factors, and weed was subjected to one-way ANOVA using pendimethalin rate as a factor at a probability level of 0.05. Treatment means were separated using Tukey's post hoc test at 95% confidence interval for significant treatment effects. All the analyses were performed in R (Russell, 2020).

## 3. Results

### 3.1. Efficacy of Different Pendimethalin Rates on Weed Establishment, Leaf Area and Dry Biomass of Barnyard Grass

There was no significant effect of different pre-emergence rates on the germination of barnyard grass (Figure 2). However, the weed establishment count of barnyard grass was lowest with 2.0 kg a.i. ha$^{-1}$ pendimethalin rate, followed by 1.75 kg a.i. ha$^{-1}$ and 1.5 kg a.i. ha$^{-1}$ S-metolachlor. The leaf area of barnyard grass significantly decreased with pendimethalin application at 40 DAS compared with the non-treated control, which reduced by 37.9, 33.5, 33.5 and 35.0% with 1.5, 1.75 and 2.0 kg a.i. ha$^{-1}$ pendimethalin and 1.5 kg a.i. ha$^{-1}$ S-metolachlor treatments, respectively (Figure 3a). At 55 DAS, only the application of 2.0 kg a.i. ha$^{-1}$ pendimethalin significantly differed in leaf area (35.1% reduction) of barnyard compared with the untreated pots (Figure 3b). Overall, there was no significant difference in mean leaf area of barnyard grass with pendimethalin rates and 1.5 kg a.i. ha$^{-1}$ S-metolachlor at 40 DAS and 55 DAS (Figure 3a,b).

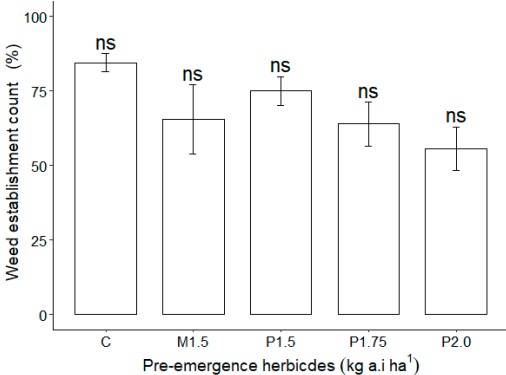

**Figure 2.** Effect of pendimethalin rates (P1.5, P1.75, P2.0 kg a.i. ha$^{-1}$), treated control with S-metolachlor at 1.5 kg a.i. ha$^{-1}$ (M1.5) and untreated control (C) on barnyard grass. 'ns' means not significant; error bars represent the standard error.

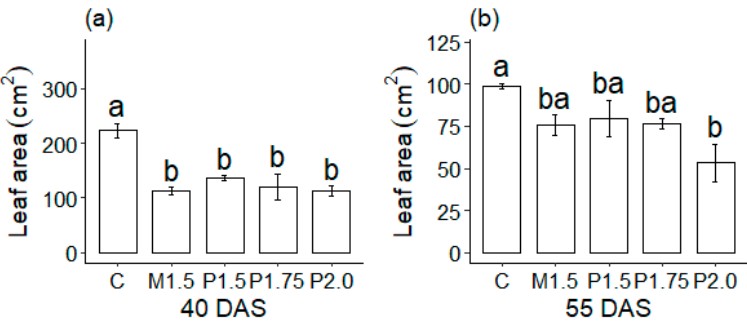

**Figure 3.** Effect of pendimethalin rates (P1.5, P1.75, P2.0 kg a.i. ha$^{-1}$), treated control with S-metolachlor at 1.5 kg a.i. ha$^{-1}$ (M1.5) and untreated control (C) on barnyard leaf area at 40 DAS (**a**) and 55 DAS (**b**). 'ns' means not significant, while means of the same letters are not significantly different at $p < 0.05$; error bars represent the standard error.

The herbicide application rates influenced leaf biomass of barnyard grass at only 40 DAS (Table 1). However, leaf biomass was significantly reduced with pendimethalin rates and 1.5 L a.i. ha$^{-1}$ S-metolachlor compared with non-treated pots. The leaf biomass of barnyard grass decreased correspondingly by 38.4, 37.1 and 36.2 with 1.5, 1.75 and 2.0 kg a.i. ha$^{-1}$ pendimethalin applications, respectively, compared with the untreated control, but pendimethalin rates and 1.5 kg a.i. ha$^{-1}$ S-metolachlor did not differ significantly in the leaf biomass of barnyard grass (Table 1). Pre-emergence herbicide rates did not affect the shoot biomass of barnyard grass at 40 DAS, unlike at 55 DAS (Table 1). At 55 DAS, 2.0 kg a.i. ha$^{1}$ pendimethalin rate significantly differed in mean shoot biomass

of barnyard grass compared with the two controls. The shoot biomass of barnyard grass was reduced by 45.6% with the highest pendimethalin (2.0 kg a.i. ha$^{-1}$) treatment over the non-treated control at 55 DAS. More so, lower pendimethalin treatments produced similar shoot biomasses of barnyard grass as the two controls. The non-treated control pots produced the highest shoot biomass up to 0.79 g plant$^{-1}$ followed by treated control (1.5 kg a.i. ha$^{1}$ S-metolachlor) 0.78 g plant$^{-1}$ at 55 DAS (Table 1).

**Table 1.** Effects of pre-emergence herbicide on leaf, shoot, above-ground and root biomass of barnyard grass.

| Pendimethalin Rates (kg a.i. ha$^{-1}$) | Leaf Biomass 40 DAS 55 DAS (g plant$^{-1}$) | | Shoot Biomass 40 DAS 55 DAS (g plant$^{-1}$) | | Above-Ground Biomass 40 DAS 55 DAS (g plant$^{-1}$) | | Root Biomass (g plant$^{-1}$) |
|---|---|---|---|---|---|---|---|
| 1.5 | 0.56 b | 0.23 a | 1.12 a | 0.54 ba | 1.68 ba | 0.82 ba | 0.42 a |
| 1.75 | 0.53 b | 0.23 a | 1.10 a | 0.53 ba | 1.63 ba | 0.77 ba | 0.38 a |
| 2.0 | 0.51 b | 0.17 a | 0.98 a | 0.43 b | 1.48 b | 0.59 b | 0.27 a |
| Control 1 | 0.90 a | 0.31 a | 1.58 a | 0.79 a | 2.47 a | 1.09 a | 0.49 a |
| M 1.5 (Control 2) | 0.45 b | 0.27 a | 0.88 a | 0.78 a | 1.33 b | 1.05 a | 0.26 a |
| p values (p < 0.05) | 0.0023 | ns | ns | 0.018 | 0.016 | 0.0083 | ns |

'Control 1' represents the untreated pots with no pendimethalin, and M1.5 is S-metolachlor applied at 1.5 kg a.i. ha$^{-1}$ as control 2 or the treated control. 'ns' means not significant, while means of the same letters are not significantly different at $p < 0.05$; error bars represent the standard error.

All the treated pots decreased total biomass at 40 and 55 DAS. Application of 2.0 kg a.i. ha$^{-1}$ pendimethalin significantly decreased the above-ground weed biomass to 1.48 g plant$^{-1}$ and by 40.1% compared with the control at 40 DAS. However, the treated control applied as 1.5 kg a.i. ha$^{-1}$ S-metolachlor significantly decreased the mean above-ground biomass by 46.2% (1.33 g plant$^{-1}$) compared with the untreated control (Table 1). There was a slight decrease in mean above-ground biomass across all the treatments at 55 DAS. Except the treated control, pendimethalin rates reduced total biomass of barnyard grass at 55 DAS compared with the non-treated control. Different pendimethalin rates did not influence root biomass. However, root biomass of barnyard grass showed a reduction trend with 0.26 g plant$^{-1}$ and 0.27 g plant$^{-1}$ when 1.5 kg a.i. ha$^{-1}$ S-metolachlor and 2.0 kg a.i. ha$^{-1}$ pendimethalin were applied, respectively (Table 1).

### 3.2. Rice Seed Germination, Leaf Area, Leaf, Shoot and Total Biomass

Application of different pendimethalin rates did not significantly affect rice germination in Petri dishes in the cabinet and pots in the greenhouse (Table 2). However, rice exposed to 2.0 kg a.i. ha$^{-1}$ pendimethalin rate in the Petri dishes reduced germination at a rate of 35% compared with the controls (Table 2). Rice seeds that received 1.5 kg a.i. ha$^{-1}$ S-metolachlor (treated control) showed a high germination with a rate more than 70%. Each herbicide rate displayed over 80% rice germination rates in pots in the greenhouse as well as the untreated control. There was no varietal difference in germination in both growth media. (Table 2).

The pendimethalin-treated plants significantly differed in leaf biomass, shoot biomass and total biomass except the total leaf area (Table 3). The treated control applied as 1.5 kg a.i. ha$^{-1}$ S-metolachlor caused more injury to rice biomass and reduced the shoot biomass by up to 0.16 g plant$^{-1}$ (i.e., by 38.5%). The leaf biomass, shoot biomass and total biomass reduction with 1.5 kg a.i. ha$^{-1}$ S-metolachlor over the untreated control was the lowest, but 1.5 kg a.i. ha$^{-1}$ S-metolachlor and pendimethalin treatments did not significantly differ in mean leaf biomass, shoot biomass and total biomass (Table 3). For varietal response, crop injury affected *Topaz* more than *Sen pidao,* and the leaf area, leaf biomass, shoot biomass and total biomass of *Topaz* significantly reduced by up to

76.9 cm$^2$ plant$^{-1}$, 0.22, 0.18 and 0.40 g plant$^{-1}$ (by 28.7, 31.3, 30.8 and 29.8%), respectively (Table 3).

**Table 2.** Effects of pre-emergence herbicides and varieties in rice germination under different growing conditions for the direct sowing method.

| Pendimethalin (P) Rates (kg a.i. ha$^{-1}$) | Germination in Petri Dishes (%) | Germination in Pots (%) |
|---|---|---|
| P 1.5 | 67.5 a | 87.5 a |
| P1.75 | 53.8 a | 81.9 a |
| P 2.0 | 35.0 a | 80.6 a |
| Control 1 | 75.0 a | 95.8 a |
| M 1.5 (Control 2) | 73.8 a | 81.9 a |
| *p* values (*p* < 0.05) | ns | ns |
| Varieties | | |
| *Topaz* | 59 a | 84.4 a |
| *Sen pidao* | 63 a | 86.7 a |
| *p* values (*p* < 0.05) | ns | ns |

'Control 1' represents the untreated pots with no pendimethalin, and M 1.5 is *S*-metolachlor applied at 1.5 kg a.i. ha$^{-1}$ as control 2 with herbicide application. 'ns' represents not significant, while means of the same letters are not significantly different at *p* < 0.05.

**Table 3.** Effect of pre-emergence herbicide and varieties on the growth of the transplanted rice seedlings.

| Pendimethalin (P) Rates (kg a.i. ha$^{-1}$) | Total Leaf Area (cm$^2$ plant$^{-1}$) | Leaf Biomass (g plant$^{-1}$) | Shoot Biomass (g plant$^{-1}$) | Total Biomass (g plant$^{-1}$) |
|---|---|---|---|---|
| P 1.5 | 99.8 a | 0.29 a | 0.24 a | 0.53 a |
| P 1.75 | 98.0 a | 0.27 ba | 0.22 ba | 0.49 a |
| P 2.0 | 78.7 a | 0.26 ba | 0.21 ba | 0.46 ba |
| Control 1 | 102.1 a | 0.32 a | 0.26 a | 0.58 a |
| M 1.5 (Control 2) | 83.2 a | 0.20 b | 0.16 b | 0.37 b |
| *p* values (*p* < 0.05) | ns | 0.0012 | 0.005 | $2.00 \times 10^{-4}$ |
| Varieties | | | | |
| *Topaz* | 76.9 b | 0.22 b | 0.18 b | 0.40 b |
| *Sen pidao* | 107.9 a | 0.32 a | 0.26 a | 0.57 a |
| *p* values (*p* < 0.05) | 0.0004 | 0.0001 | 0.0001 | 0.0001 |

'Control 1' represents the untreated pots with no pendimethalin, and M 1.5 is *S*-metolachlor applied at 1.5 kg a.i. ha$^{-1}$ as control 2 with herbicide application. 'ns' represents not significant, while means of the same letters are not significantly different at *p* < 0.05.

*3.3. Varietal Responses to Pendimethalin Rates on SPAD, and Maximum Quantum Yield and Effective Quantum Yield of Photosystem II*

Application of pendimethalin rates significantly interfered with SPAD values for thed *Topaz* variety only at 7 DAT (Figure 4a). The SPAD values decreased up to 36.1 (13%) and 34 (18%), with 2.0 kg a.i. ha$^{-1}$ pendimethalin rates and 1.5 kg a.i. ha$^{-1}$ *S*-metolachlor, respectively (Figure 4a). The lowest (1.5 kg a.i. ha$^{-1}$) pendimethalin treatment significantly increased the highest mean SPAD value (40.4) compared with 2.0 kg a.i. ha$^{-1}$ pendimethalin. Pendimethalin treatments did not affect SPAD values of the *Sen pidao* variety at 7 and 12 DAT; however, 1.5 kg a.i. ha$^{-1}$ *S*-metolachlor had the lowest SPAD value at these growth stages (Figure 4c,d).

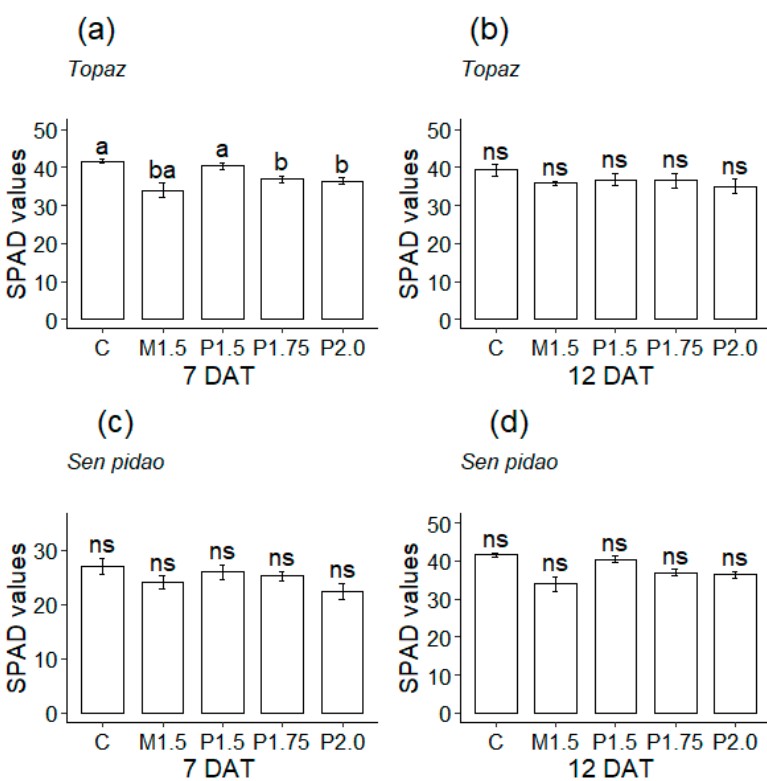

**Figure 4.** SPAD or leaf greenness as affected by pendimethalin rates as P 1.5, 1.75, 2.0 kg a.i. ha$^{-1}$ and C as the untreated control and M 1.5 as control 2 with 1.5 kg a.i. ha$^{-1}$ *S*-metolachlor herbicide for *Topaz* variety at 7 DAT (**a,b**) and *Sen pidao* variety at 12 DAT (**c,d**). 'ns' represents not significant, while means of the same letters are not significantly different at $p < 0.05$; error bars represent the standard error.

There were significant varietal responses to pendimethalin rates on maximum quantum yield (dark-adapted Fv/Fm) and effective quantum yield (light-adapted F'v/F'm) of photosystem (PS) II except for maximum quantum yield that was estimated in the *Sen pidao* variety (Figure 5). The mean effective quantum yield F'v/F'm of the *Topaz* variety was more than 0.6 with lower pendimethalin rates. For the *Topaz* variety, the effective quantum yield significantly decreased by over 15.9% (0.578) with 2.0 kg a.i. ha$^{-1}$ pendimethalin compared with the untreated controls (Figure 5b). However, lower pendimethalin rates and 1.5 kg a.i. ha$^{-1}$ *S*-metolachlor treatments did not significantly differ in the effective quantum yield of the *Topaz* variety. The effective quantum yield of the *Topaz* variety increased correspondingly up to 0.65, 0.64 and 0.63 with 1.5, 1.75 kg a.i. ha$^{-1}$ pendimethalin rates and 1.5 kg a.i. ha$^{-1}$ s-metolachlor, respectively (Figure 5b).

There was a significant ($p < 0.05$) effect of pendimethalin rates on maximum quantum yield (dark-adapted) and effective quantum yield of *Sen pidao* (Figure 5c,d). The maximum quantum yield of the *Sen pidao* variety significantly decreased by 19% with 2.0 kg a.i. ha$^{-1}$ pendimethalin compared with the untreated control but was similar to the treated control. However, different pendimethalin rates did not vary in the maximum quantum yield of the *Sen pidao* variety (Figure 5c).

Compared with untreated control, the effective quantum yield (light-adapted F'v/F'm) of the *Sen pidao* variety significantly decreased up to 0.52 when 2.0 kg a.i. ha$^{-1}$ pendimethalin was applied (Figure 5d). Whereas, the mean effective quantum yield of *Sen pidao* was above 0.6 with 1.5 kg a.i. ha$^{-1}$ pendimethalin rates as the control. The application of 1.75 kg a.i. ha$^{-1}$ pendimethalin and treated control also reduced the effective quantum yield of the *Sen pidao* variety.

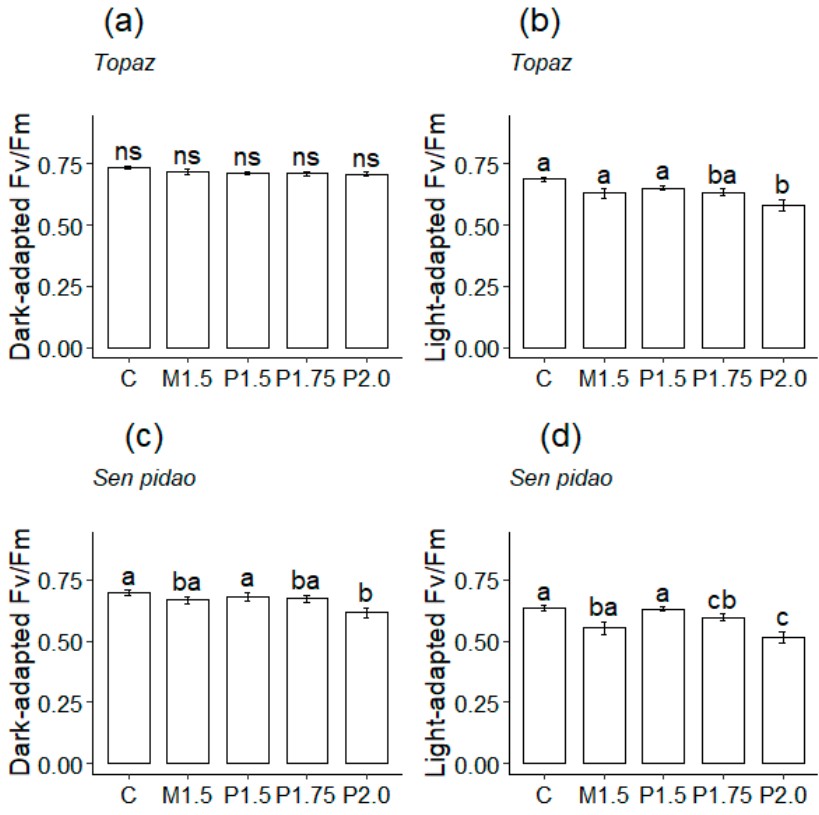

**Figure 5.** Dark-adapted (Fv/Fm) and light-adapted (F'v/F'm) of *Topaz* (**a**,**b**) and *Sen pidao* (**c**,**d**) varieties, respectively, under different herbicide treatments; P1.5, P1.75, P2.0 as pendimethalin rates at 1.5, 1.75, 2.0 kg a.i. ha$^{-1}$, respectively, C represents the untreated control and M 1.5 is control 2 with 1.5 kg a.i. ha$^{-1}$ *S*-metolachlor herbicide. 'ns' represents not significant, while means of the same letters are not significantly different at $p < 0.05$; error bars represent the standard error.

## 4. Discussion

### 4.1. Growth Attributes and Biomass of Barnyard Grass

This study demonstrated that all pendimethalin rates and the treated control showed high efficacy in controlling total biomass of barnyard grass, and no appreciable differences occurred in controlling leaf area and leaf shoot biomass of barnyard grass. The similar impact on the reduction in the total biomass of barnyard grass from varying pendimethalin herbicides could be associated with a high herbicide solubility that might have resulted in the absorption of pendimethalin by the root system of barnyard grass (Table 1). It is not clear why there was no significant effect of different pendimethalin treatments on seed germination of barnyard grass; meanwhile, more reduction in the leaf area at 40 and 55 DAS at different pendimethalin rates occurred. To begin with, this is possible because the remaining barnyard grass seeds germinated under a thin chemical barrier of adsorbed herbicides on the outer surface of seed coats in the enclosed pots, indicating that the accumulated pre-emergence herbicides in the soil interacted with the germinated seeds and eventually became absorbed through the roots. In addition, herbicides applied on the soil surface are adsorbed by the soil colloids and form a shallow herbicidal layer, through which the new shoots of germinating weeds can absorb the chemical to show some phytotoxic symptoms. A past study reported more than 10% barnyard grass germination and seedling emergence with 1 and 2 kg a.i. ha$^{-1}$ pendimethalin application [26]. A different study also reported that pendimethalin rates applied as low as 149, 298, 596 and 1192 g a.i. ha$^{-1}$ provided 100% control of *E. colona* [27]. This study indicates that lower pendimethalin rates and 1.5 kg a.i. ha$^{-1}$ *S*-metolaclor are relatively economical to provide excellent control of barnyard grass rather than higher rates of pendimethalin (Table 1). These results suggest

that 1.5 kg a.i. ha$^{-1}$ rate pendimethalin application is effective to control barnyard grass and to promote high crop competition under field conditions.

### 4.2. Rice Germination and Growth

This study showed no rice germination response from different pendimethalin rates with Petri dishes and pots for the direct sowing method (Table 3). The 38.5% (0.18 g plant$^{-1}$) shoot biomass reduction in rice with 1.5 kg a.i. ha$^{-1}$ S-metolachlor implies that 1.5 kg a.i. ha$^{-1}$ S-metolachlor was more toxic and easily taken up by rice plants at a lower rate than 1.5 kg a.i. ha$^{-1}$ pendimethalin (Table 3). A past study reported up to 36% rice injury when 1.42 kg ai ha$^{-1}$ S-metolachlor was applied in the rice field [28]. The information generated in this study show that 1.5 kg a.i. ha$^{-1}$ pendimethalin herbicide was safer for rice plants due to a lower phytotoxicity effect on the rice plant. Generally, based on the rice varietal response on above-ground biomass, the results showed greater reduction in biomass from 37 to 41% for the *Topaz* variety; leaf area, leaf biomass, shoot biomass and total biomass of *Topaz* rice reduced more by up to 28.7, 31.3, 30.8 and 29.8%, respectively, than *Sen pidao*. The results of these experiments contribute to our understanding of the growth and physiological response of rice plants to different pendimethalin rates and their extent of damage in rice plants. In addition, it validates the results of the field experiments we conducted that resulted in minimal crop injury from a 1.5 kg a.i. ha$^{-1}$ pendimethalin rate (unpublished).

### 4.3. SPAD and Photosynthetic Performance of Photosystem II

Taken together are the varietal responses to pendimethalin rates on SPAD value (leaf greenness), maximum quantum yield and effective quantum yield of PS II at different growth stages. Phytotoxicity was more severe on SPAD values of the *Topaz* variety at higher rates of pendimethalin and 1.5 kg a.i. ha$^{-1}$ S-metolachlor at 7 DAT (Figure 4a). It is still not clear why there was no difference observed with pendimethalin rates on SPAD values of the *Sen pidao* variety 7 and 14 DAT considering that the experiment was conducted under the same environmental conditions. The explanation of low SPAD values (>30) of the *Topaz* variety with higher pendimethalin rates and 1.5 kg a.i. ha$^{-1}$ S-metolachlor might be due to impairment or weakness in plant cells leading to changes in chlorophyll pigments and leaf discolouration in plants. Herbicides taken up by plants can influence cell metabolism and cause specific alterations in photosynthetic capacity and chlorophyll pigment [29].

In addition, the greater reduction in the effective quantum yield of PS II of *Sen pidao* and *Topaz* varieties from 2 kg a.i. ha$^{-1}$ pendimethalin herbicide could be attributable to a greater uptake of reactive oxygen species (ROS) and activated high oxidative stress, which might inhibit electron transport in the chloroplasts and the reaction centres in photosystem II. Further investigation showed that higher pendimethalin rates inhibited maximum quantum yield (dark-adapted) and effective quantum yield (light-adapted) of photosystem II of *Sen pidao*, more than *Topaz*. Therefore, we speculate that the leaf architecture and chemical makeup of cuticles can influence the herbicidal activity and mechanisms, but this is beyond the context of our study. This suggests that 1.5 kg a.i. ha$^{-1}$ pendimethalin was safer in growing *Sen pidao* and *Topaz* varieties.

### 5. Conclusions

All pendimethalin rates and S-metolachlor provided excellent control for barnyard grass. Higher pendimethalin and 1.5 kg a.i. ha$^{-1}$ S-metolachlor interfered with maximum quantum yield and effective quantum yield of photosystem II *Sen pidao* variety and caused greater toxicity. The reduction in leaf greenness and total biomass of the *Topaz* variety may be compensated using better management practices which may be explored in some related work. This information generated from the photosynthetic parameter of the light-adapted technique can still be used by the rice growers in the field to characterise rice varieties from herbicide toxicity. Therefore, an application of 1.5 kg a.i. ha$^{-1}$ pendimethalin was effective to control barnyard grass and was safer and more economical for rice production.

Although the transplanting method used in this study is not a replication of standard farmer's practice for DSR, it appears that this sowing technique would lend itself well for use specifically for greenhouse experiments performed by plant breeders and scientists.

**Author Contributions:** Conceptualisation, C.B.O.-H., F.V.O. and D.K.Y.T.; methodology, C.B.O.-H., F.V.O., D.K.Y.T. and R.C.; statistical analysis, C.B.O.-H., F.V.O., D.K.Y.T. and R.C.; investigation and data processing, C.B.O.-H., F.V.O., D.K.Y.T. and R.C.; writing of the original manuscript, C.B.O.-H.; review and editing, D.K.Y.T., F.V.O. and R.R.; supervision, D.K.Y.T., F.V.O. and R.R. All authors have read and agreed to the published version of the manuscript.

**Funding:** This research was funded by the Australian Centre for International Agricultural Research (ACIAR) Project number CSE/2015/044. The Postgraduate Internship was funded by the Australian Plant Phenomics Facility (APPF) in CSIRO.

**Institutional Review Board Statement:** Not Applicable.

**Informed Consent Statement:** Not Applicable.

**Data Availability Statement:** Data are available on request.

**Acknowledgments:** We thank the Australian Centre for International Agricultural Research (ACIAR) Project number CSE/2015/044 for funding this research and Tertiary Education Trust Fund (TETfund), CSIRO/DATA 61 for the financial assistance on the field expenses and the APPF team in Canberra for their technical support. We thank Robert Martin, Ratha Rien, Sophea Yous and Chariya Korn for assistance in the fieldwork in northwest Cambodia. This manuscript is part of a chapter submitted as partial fulfillment of a Doctor of Philosophy (PhD) degree at the University of Sydney.

**Conflicts of Interest:** The authors declare no conflict of interest.

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
