# Peer review of "Efficacy of Pendimethalin Rates on Barnyard Grass (Echinochloa crus-galli (L.) Beauv) and Their Effect on Photosynthetic Performance in Rice"

_agronomy, doi:10.3390/agronomy13020582_

Round 1

Reviewer 1 Report

This paper investigated the effects of pendimethalin application to barnyard grass and rice. Overall this is a poorly written manuscript, with lots of typos, extra space and wrong units etc. If you check the herbicide label of pendimethalin in Australia, the recommended rate to use in the rice field never exceeds 1.5 L a.i. ha-1. Therefore I do not see a big point of this research.  

Line 12. Give the full name of DSR.

Line 15 17.5 Should it be “1.75 L a.i. ha-1”. Check the whole manuscript.

Line 23 “S”-metolachlor, the “S” should be capital. Check the whole manuscript including the legend.

Line 26. There is an extra space within 0.58

Line 33. No period.

Line 57. Reference 10 did not mention CO2 fixation.

Line 61. Reference 12 did not mention geranylgeranyl pyrophosphate cyclisation enzymes as the mechanisms of S-metolachlor.

Line 66. Neither 13 nor 14 references mentioned pendimethalin exposure to plants potentially increase physiological changes including the PSII performance.

Line 105. No info about barnyard grass.

Line 138. Please specify the volume unit of pendimethalin

Line 192. What does it mean by “germination counts”? I cannot see “considerably reduction” from figure 2.

Line 204 What it means by “no mean difference”?

Line 338 the unit of the herbicide rate is wrong. 

Author Response

Response to Reviewer 1 comments

This paper investigated the effects of pendimethalin application to barnyard grass and rice. Overall this is a poorly written manuscript, with lots of typos, extra space and wrong units etc. If you check the herbicide label of pendimethalin in Australia, the recommended rate to use in the rice field never exceeds 1.5 L a.i. ha-1. Therefore I do not see a big point of this research.

Response: You are correct about the recommended pendimethalin rate (1.5 kg a.i. ha-1) in Australia. However, we exceeded it to obtain the upper limit of rice growth and physiological response with different pendimethalin rates in relation to herbicide damage. This information is useful in the rice-growing regions in developing countries where herbicides are not applied with precision application technologies to minimise herbicide drift in the field.

Again, it was not evident in our previous study whether 1.5 kg a.i ha-1 pendimethalin herbicide resulted in crop injury in the farmers’ fields due to environmental and seasonal variations. So, these pot experiments on efficacy of herbicides in weed management increased our understanding of the growth and physiological response to different pendimethalin rates in direct-seeded rice (DSR). (page 11, lines 365 to 369) 

Point 1: Line 12. Give the full name of DSR.

Response: The changes have been made and corrected the full name to direct-seeded rice (DSR) on page 1, line 12.

Point 2: Line 15 17.5 Should it be “1.75 L a.i. ha-1”. Check the whole manuscript.

Response: We have made the changes and corrected to 1.75 kg a.i ha-1 on page 1 line 15.

Point 3: Line 23 “S”-metolachlor, the “S” should be capital. Check the whole manuscript including the legend.

Response: We have made the changes and corrected to S-metolachlor in the text (pages 2 to 11, lines 16, 19, 34….).

Point 4: Line 26. There is an extra space within 0.58

Response: The changes have been made on page 1 line 26.

Point 5: Line 33. No period.

Response: The changes have been made on page 1 line 36.

Point 6: Line 57. Reference 10 did not mention CO2 fixation.

Response: The changes have been made and replaced with the correct reference on page 2 line 60.

Point 7: Line 61. Reference 12 did not mention geranylgeranyl pyrophosphate cyclisation enzymes as the mechanisms of S-metolachlor.

Response: The changes have been made and replaced with the correct reference on page 2 line 64.

Point 8: Line 66. Neither 13 nor 14 references mentioned pendimethalin exposure to plants potentially increase physiological changes including the PSII performance.

Response: The changes have been made and replaced with the correct reference on page 2 line 66.

Point 9: Line 105. No info about barnyard grass.

Response: The information where the seeds of barnyard grass and rice varieties were collected are well detailed (lines 105-117), including details on barnyard experiments, treatments and the experimental layout (lines 118 to 126).

Point 10: Line 138. Please specify the volume unit of pendimethalin

Response: The volume unit has been specified and corrected to 500 µL in 50 mL on page 4 line 137.

Point 11: Line 192. What does it mean by “germination counts”? I cannot see “considerably reduction” from figure 2.

Response:  The germination count means weed establishment count. The change has been made and the sentence has been rephrased to “the weed establishment count of barnyard grass was lowest with 2.0 kg a.i ha-1 pendimethalin rate, followed by 1.75 kg a.i ha-1 and 1.5 kg a.i ha-1 S-metolachlor herbicides” (page 5, lines 194 to 196).

Point 12: Line 204 what it means by “no mean difference”?

Response: The error has been corrected and we have made the changes based on the data presented in Table 1 and replaced with “The leaf biomass of barnyard grass decreased correspondingly by 38.4, 37.1, and 36.2 with 1.5 1.75 and 2.0 kg a.i ha-1 pendimethalin application, respectively, compared with the untreated control, but pendimethalin rates and 1.5 kg a.i ha-1 S-metolachlor did not differ significantly in leaf biomass of barnyard grass (page 5, lines 206 to 210).

Point 13: Line 338 the unit of the herbicide rate is wrong.

Response: The changes have been made and the herbicide unit corrected to kg a.i ha-1 (page 12, line 352).

Reviewer 2 Report

This study was performed to evaluate the appropriate efficacy of pendimethalin in rice to control barnyard grass. Reading this manuscript, I felt that the editing of the entire text was not thoroughly done by the first and corresponding authors. Especially, English should be re-checked in terms of typos and inappropriate notation. Additionally, some of the terminology, notation, and unit associated with the chemistry field are not correct. The authors need to read the overall manuscript again and correct it. I leave some major/minor comments below.

1. The botanical nomenclature of barnyardgrass must include the authority (ex. Echinochloa crus-galli (L.) Beauv.).

2. Line 12: You haven't abbreviated DSR before.

3. Line 15: Active ingredients (a.i.) of herbicides are not liquid phases. The amount of a.i. should be mg, g, or kg unit.

4. Line 16: Correct "S-Metolachlor" to "S-metolachlor".

5. Line 19: "s-metolachlor" is a misnomer. The "S" should be italicized and capitalized since it means a levo-rotary enantiomer of metolachlor. This needs to be applied to the entire text.

6. Lines 29-31: Expressions are ambiguous. More detailed comparisons are needed. In all sentences, compare results as numerics to show how much was suppressed/inhibited.

7. Lines 32-33: This study is not a field study. Can you say this conclusion using only pot and Petri-dish experiments?

8. Line 86: I am not sure if SPAD was abbreviated before.

9. Line 199: Why are you abbreviating the weed name to "barnyard"? This is also shown in other parts of the text.

10. In the entire text, there are many parts where a space bar was clicked twice.

Author Response

Point 1: The botanical nomenclature of barnyard grass must include the authority (ex. Echinochloa crus-galli (L.) Beauv.).

Response: The correct botanical nomenclature of barnyard grass has been replaced with Echinochloa crus-galli (L.) Beauv (page 1- line 3; page 2 - lines 41 and 53, page 3-lines 98, 105….).

Point 2: Line 12: You haven't abbreviated DSR before.

Responses: We have made the changes and written the full name as direct seeded rice (DSR) (page 1, line 12).

Point 3: Line 15: Active ingredients (a.i.) of herbicides are not liquid phases. The amount of a.i. should be mg, g, or kg unit.

Responses: We have made the changes and replaced the amount of active ingredients (a.i.) of herbicides with “kg” in the text (pages 1 to 12, lines 15, 19….).

Point 4: Line 16: Correct "S-Metolachlor" to "S-metolachlor".

Response: The spelling of s-metolachlor has been corrected and replaced with S-metolachlor (page 1, lines 16 and 20; page 2, lines 61 and 62; page 3, lines 123 and133; page 5, lines 196, 199, 203, 206, 209...).

Point 5: Line 19: "s-metolachlor" is a misnomer. The "S" should be italicized and capitalized since it means a levo-rotary enantiomer of metolachlor. This needs to be applied to the entire text.

Response: We have made the changes and italicised S-metolachlor with the capital letter (page 1, line 20; pages 2 to 12).

Point 6: Lines 29-31: Expressions are ambiguous. More detailed comparisons are needed. In all sentences, compare results as numerics to show how much was suppressed/inhibited.

Response: The changes have been made and we corrected them to

“All pendimethalin treatments suppressed above-ground biomass at 55 DAS, but above-ground biomass of barnyard grass was significantly decreased by 59.9% when 2.0 kg a.i ha-1 pendimethalin was applied compared with the untreated control. Although the application of 1.5 kg a.i ha-1 pendimethalin rates reduced the effective quantum yield (light–adapted) of photosystem II of Sen pidao (0.55) by 8% compared with Topaz (0.65), low (1.5 kg a.i ha-1) pendimethalin was non-toxic for both varieties compared with 2.0 kg a.i ha-1 pendimethalin” on page 1 lines 29 to 34.

Point 7: Lines 32-33: This study is not a field study. Can you say this conclusion using only pot and Petri-dish experiments?

Response: This experiment was conducted to validate the results of our previous field study on pendimethalin toxicity in rice plants (unpublished). For the field experiments, it was not clear whether the 1.5 kg a.i ha-1 pendimethalin herbicide we applied or herbicide drift and environmental/seasonal variations resulted in crop injury during the cropping seasons. So, this further investigation on pendimethalin toxicity in rice plants in controlled environments was to increase our understanding of the growth and physiological response to different pendimethalin rates and their extent of damage in rice plants at the early growth stage (page 11, lines 365 to 369).

Point 8: Line 86: I am not sure if SPAD was abbreviated before.

Response: Yes, we have written the full meaning as single-photon avalanche diode (SPAD) on page 2 line 87.

Point 9: Line 199: Why are you abbreviating the weed name to "barnyard"? This is also shown in other parts of the text.

Response: We have corrected it to barnyard grass on page 5 line 202.

Point 10: In the entire text, there are many parts where a space bar was clicked twice.

Response: We have applied uniform spaces in the entire text.

Reviewer 3 Report

Dear Editor,

This paper investigates the efficacy of pendimethalin rates on Barnyard grass (Echinochloa crus-galli) and their effect on photosynthetic performance in rice.

The work is technically well done, with sufficient analysis. Also, the subject is interesting, and the study was well-designed and performed.

The manuscript is recommended for publication in Agronomy Journal with a few minor comments:

Line 2: Please, in Title correct Echinochloa crus-galli to Echinochloa crus-galli.

Lines 16, 19.... : Please, make the spelling of s-metolachlor consistent throughout the text.

Lines 17 and 379: Please, italicize Topaz.

Author Response

Response to comments (Reviewer 3)

This paper investigates the efficacy of pendimethalin rates on Barnyard grass (Echinochloa crus-galli) and their effect on photosynthetic performance in rice. The work is technically well done, with sufficient analysis. Also, the subject is interesting, and the study was well-designed and performed. The manuscript is recommended for publication in Agronomy Journal with a few minor comments:

Point 1: Line 2 Please, in Title correct Echinochloa crus-galli to Echinochloa crus-galli.

Response: We have made the changes and corrected it to Echinochloa crus-galli (L.) Beauv (page 1- line 3; page 2 - lines 41 and 53, page 3-lines 98, 105…).

Point 2: Lines 16, 19.... : Please, make the spelling of s-metolachlor consistent throughout the text.

Response: The spelling of s-metolachlor has been corrected and replaced with S-metolachlor on page 1, lines 16 and 20; page 2, lines 61 and 62; page 3, lines 123 and133; page 5, lines 196, 199, 203, 206, 209...

Point 3: Lines 17 and 379: Please, italicize Topaz.

Response: We have made the changes and corrected it to Topaz on page 1 line 18 and page 18 line 397.
